# A Viral Suppressor of RNA Silencing May Be Targeting a Plant Defence Pathway Involving Fibrillarin

**DOI:** 10.3390/plants11151903

**Published:** 2022-07-22

**Authors:** Miryam Pérez-Cañamás, Michael Taliansky, Carmen Hernández

**Affiliations:** 1Instituto de Biología Molecular y Celular de Plantas (Consejo Superior de Investigaciones Científicas-Universidad Politécnica de Valencia), Ciudad Politécnica de la Innovación, Calle Ingeniero Fausto Elio, Ed. 8E. Camino de Vera s/n, 46022 Valencia, Spain; mirpreca@etsia.upv.es; 2The James Hutton Institute, Invergowrie, Dundee DD2 5DA, UK; michael.taliansky@hutton.ac.uk; 3Shemyakin-Ovchinnikov Institute of Bioorganic Chemistry of the Russian Academy of Sciences, 117997 Moscow, Russia

**Keywords:** viral suppressor of RNA silencing, fibrillarin, plant RNA virus, antiviral defence, nucleolus, Cajal body, coilin, plant–virus interactions, *Nicotiana benthamiana*, PLPV

## Abstract

To establish productive infections, viruses must be able both to subdue the host metabolism for their own benefit and to counteract host defences. This frequently results in the establishment of viral–host protein–protein interactions that may have either proviral or antiviral functions. The study of such interactions is essential for understanding the virus–host interplay. Plant viruses with RNA genomes are typically translated, replicated, and encapsidated in the cytoplasm of infected cells. Despite this, a significant array of their encoded proteins has been reported to enter the nucleus, often showing high accumulation at subnuclear structures such as the nucleolus and/or Cajal bodies. However, the biological significance of such a distribution pattern is frequently unknown. Here, we explored whether the nucleolar/Cajal body localization of protein p37 of *Pelargonium line pattern virus* (PLPV, genus *Pelarspovirus*, family *Tombusviridae*), might be related to potential interactions with the nucleolar/Cajal body marker proteins, fibrillarin and coilin. The results revealed that p37, which has a dual role as coat protein and as suppressor of RNA silencing, a major antiviral system in plants, is able to associate with these cellular factors. Analysis of (wildtype and/or mutant) PLPV accumulation in plants with up- or downregulated levels of fibrillarin or coilin have suggested that the former might be involved in an as yet unknown antiviral pathway, which may be targeted by p37. The results suggest that the growing number of functions uncovered for fibrillarin can be wider and may prompt future investigations to unveil the plant antiviral responses in which this key nucleolar component may take part.

## 1. Introduction

Viruses are among the most devastating microorganisms throughout all kingdoms of life, posing serious threats to human and animal health and leading to huge crop losses. As obligatory intracellular parasites, their surveillance relies on a vast array of host factors that are frequently hijacked by virus components or functionally subverted. The host, on its side, deploys diverse countermeasures to prevent or limit the progress of viral infection, which, in turn, are targeted by the virus to avoid being defeated [1]. The outcome of this continuing arms race can be variable, from non-infection to a highly productive infection that, moreover, may have a significant impact on the host [2].

Viruses with RNA genomes usually replicate in the cell cytoplasm though RNA intermediates generated by the viral RNA dependent-RNA polymerase (RDR). Translation of viral proteins and encapsidation of viral progeny will also occur in this cellular compartment [3]. Despite their infectious cycle being eminently cytoplasmic, a non-negligible amount of proteins encoded by RNA viruses enters the nucleus, though the biological meaning of such behaviour is often unknown [4]. In the case of plant RNA viruses, nuclear localization has been described for a variety of viral proteins encoded by members of distinct genera/families though information on its implications for either the virus or the host is scarce. Moreover, many of these proteins reach high accumulation levels at nucleolus and Cajal bodies (CBs), two subnuclear structures that are interconnected and that have been classically involved in maturation of ribosomal RNA (rRNA) and assembly of small nuclear (sn) ribonucleoproteins (RNPs), respectively, though a growing body of evidence indicates they may participate in many other important cell functions [5,6,7,8]. One of the best studied cases of a plant viral protein accessing the nucleus concerns the movement protein (MP) encoded by an umbravirus, a coat protein (CP)-lacking virus. This MP has been reported to recruit fibrillarin, a major nucleolar/CB protein, to form RNP complexes that mediate virus long-distance transport [9]. A similar mechanism has been proposed to support helper virus-independent long-distance trafficking of a subviral satellite RNA [10]. An influence (positive or negative) of fibrillarin and, also, of coilin, the signature protein of CBs, has also been reported in several other plant–virus interactions though the mechanistic basis of that influence is not always well understood [11,12].

*Pelargonium line pattern virus* (PLPV) has a single-stranded (ss) RNA genome of plus polarity and belongs to genus *Pelarspovirus* within family *Tombusviridae* [13]. Its unique genomic RNA, of 3883 nt, contains five open reading frames (ORFs) that encode, respectively, two proteins involved in replication (p27 and its readthrough product p87, the viral RDR), two small MPs (p7 and p9.7), and a CP of 37 kDa (p37) [14,15]. This latter protein is also able to interfere with RNA silencing (also known as RNA interference or RNAi); thus, it has a dual function as CP and viral suppressor of RNA silencing (VSR) [16]. RNA silencing acts as a major antiviral defence mechanism in plants and is triggered by double-stranded (ds) RNAs that are processed by RNase III Dicer-like enzymes to generate small RNAs (sRNAs). One of the strands of these small duplexes is incorporated into an RNA-induced silencing complex (RISC) that contains an endonuclease of the Argonaute family. RISC is then directed by the associated sRNA to a target RNA of complementary sequence, promoting its degradation [17]. Viruses produce dsRNAs during their infectious cycle that serve as elicitors of RNA silencing. VSRs may hinder distinct steps of the defence pathway to prevent viral clearance [18]. Previous studies have shown that PLPV p37 binds sRNAs, hence precluding their incorporation into RISC. Strikingly, this protein localizes not only in the cytoplasm, where encapsidation and VSR functions are expected to take place, but also in the nucleus [16]. Within the nucleus, p37 showed high accumulation at the nucleolus and CBs, as was confirmed through colocalization experiments with fibrillarin. The nucleolar localization of p37 was found to be advantageous for PLPV infection, though the ultimate reasons of this phenomenon were unclear [19].

In this work, we examined whether PLPV p37 may not only colocalize with fibrillarin but interact with it as well. We also included coilin in the protein–protein interaction assays. As we have found that p37 is able to interact with both nuclear components, we have tried to obtain insights into the potential biological meaning of such interactions. Toward this objective, we analysed the progression of infections established by wildtype (wt) and VSR-defective PLPV in plants with fibrillarin or coilin content diminished through RNAi approaches. The results have suggested that fibrillarin, whose levels are not misregulated by PLPV infection, may be involved in a host antiviral defence response, which is targeted by p37. Whether such response is related to RNA silencing or to a different pathway is a question that warrants further investigation.

## 2. Results

### 2.1. P37 Interacts In Vivo with Both Fibrillarin and Coilin

As mentioned above, PLPV p37 has been previously detected in the nucleus of plant cells through both confocal laser microscopy and subcellular fractionation experiments [16,19]. In the first type of experiments, colocalization of p37 with fibrillarin corroborated the presence of the viral protein in the nucleolus and CBs, two prominent nuclear subcompartments [16]. In the light of this background, we considered it relevant to analyse whether these proteins were not only colocalizing but also interacting with each other. To asses this possibility, a bimolecular fluorescence complementation (BiFC) assay was carried out. For this purpose, constructs for transient expression of fibrillarin (FIB) fused to the N- or C-terminal part of the super yellow fluorescent protein (sYFP) were generated. These constructs were agroinfiltrated in proper combinations with others allowing transient expression of p37 fused to the N- or C-terminal part of sYFP [16]. *Nicotiana benthamiana* cells co-expressing sYFPN:p37 and sYFPC:FIB (or the opposite combination sYFPC:p37 plus sYFPN:FIB; data not shown) showed clear sYFP-derived fluorescence at 3 days post-infiltration (dpif) (Figure 1, panels A1–A4), indicating reconstitution of the fluorophore, thus demonstrating that p37 associates with fibrillarin. Moreover, such association seems to take place in the nucleus and, particularly, in the nucleolus and CB according to the fluorescence patterns (Figure 1).

Coilin (COIL) was also employed in BiFC assays with p37 given its reported modulation effects in some plant–virus interactions [11]. Constructs allowing expression of sYFPN:COIL and sYFPC:COIL were engineered and used in proper combinations with constructs sYFPN:p37 and sYFPC:p37 to agroinfiltrate plants. Confocal microscopy observations revealed clear fluorescence in leaf cells co-expressing sYFPN:p37 and sYFPC:COIL (Figure 1B1–B4) or vice versa (sYFPC:p37 and sYFPN:COIL; data not shown), substantiating the establishment of p37–coilin interactions. As observed with fibrillarin, such interactions were detected in the nucleus with strong signals in the nucleolus and CBs. Control experiments in which the distinct fusion proteins were co-expressed with unfused sYFP halves did not yield any fluorescent signal (Figure 1A5,B5), validating the detected interactions. These interactions were further validated by the employment of additional control BiFC constructs directing expression of the two transcription factors, FUL and SOC, which did not give rise to any fluorescent signal in combination with the p37 construct (Figure 1C1–C4,D1–D4). In contrast, the interaction between FUL and SOC was evident (Figure 1E1–E4), in agreement with previous results [20].

Collectively, the obtained data showed that PLPV p37 is able to interact with two signature protein components of the nucleolus and CBs, fibrillarin and coilin, in the cellular subcompartments where these proteins preferentially accumulate. Strikingly, interaction of p37 with coilin was detected not only in the CBs, where coilin acts as a scaffolding component, but also in the nucleolus, suggesting a p37-mediated relocalization of the protein. Alternatively, overexpression of coilin, driven by a 35S promoter, as in the present case, may result in some accumulation of the protein at nucleolus, as has been noted in previous reports [11,21].

### 2.2. Downregulation of Fibrillarin but Not Coilin Results in Heightened Accumulation of an VSR Defective-PLPV

The results obtained in the previous section prompted us to explore whether the newly identified interacting partners of p37 may have an influence on PLPV infection. For this purpose, *N. benthamiana* lines in which the expression levels of fibrillarin and coilin genes were separately downregulated through RNAi approaches (hereafter, FIBi and COILi lines, respectively) [10,11] were challenged with the wt virus. Non-modified (wt) *N. benthamiana* plants were also inoculated with wt PLPV for comparison. Harvesting of inoculated leaves at 7 days post-inoculation (dpi) and analysis of the accumulation levels of PLPV through Northern blot hybridization showed a slight augmentation of viral titres in FIBi plants (Figure 2A, left), though further assessment through reverse transcription-quantitative PCR (RT-qPCR) did not support significant differences among the distinct plant genotypes (Figure 2A, right).

In a previous work, the impact of the impairment of components of the RNA silencing machinery on PLPV infection was particularly obvious when a VSR-defective virus was employed for the bioassays [22]. This was in agreement with that observed in other plant–virus interactions (e.g., [23,24]) and is due to the fact that VSRs may mask the effect of host components involved in the host defence against a virus. Due to the scarcity of data available on this topic, we considered the possibility that fibrillarin and/or coilin could participate in antiviral responses that might be targeted by p37. To test this possibility, wt, FIBi, and COILi *N. benthamiana* plants were inoculated with a VSR-deficient PLPV mutant, PLPV-mutp37WA. This mutant harbours a W28A amino acid substitution in p37 that abolished its VSR activity but preserved its encapsidation function [16]. As shown previously [16], this p37 engineered variant (p37WA) did not maintain the nucleolar/CB localization. Interestingly, a BiFC assay indicated that, in contrast with wt p37, it was unable to interact with either fibrillarin or coilin (Figure 3).

Northern blot analysis of leaves inoculated with PLPV-mutp37WA at 7 dpi showed an obvious increment in viral RNA accumulation in FIBi *N. benthamiana* plants when compared with wt *N. benthamiana* plants (Figure 4), where such accumulation was barely detectable, in agreement with earlier observations [16,22]. The accumulation levels of the mutant virus in COILi *N. benthamiana* were also very low and comparable with those found in wt *N. benthamiana* (Figure 4). In contrast with the wt virus (Figure 1), PLPV-mutp37WA did not become systemic in any plant genotype (data not shown), paralleling what has been reported in *N. benthamiana* lines impaired in RNA silencing components [22]. Altogether, the results supported that fibrillarin is involved in a PLPV antiviral defence that may be interfered by p37, but not by p37WA. The lack of interaction of the mutated protein with fibrillarin was further confirmed by a co-immunoprecipitation assay (Figure 5).

### 2.3. Overexpression of Fibrillarin Has a Negative Effect on PLPV Accumulation

Taking into consideration the positive impact that fibrillarin knockdown had on PLPV accumulation, particularly noticeable with the VSR-defective virus (Figure 4), we examined whether overexpression of this nucleolar protein could lead to the opposite outcome, i.e., a reduction in viral accumulation. We tested this possibility with the wt PLPV since the titres of the PLPV-mutp37WA were virtually imperceptible in wt *N. benthamiana* plants (Figure 4, lanes 4–5). To approach this issue, FIB:mRFP was transiently expressed in wt *N. benthamiana* leaves through agroinfiltration and, at 3 dpif, the leaves were challenged with wt PLPV. Three days later, a significant reduction in virus accumulation levels was detected in FIB:mRFP overexpressing leaves in comparison with the mock controls (Figure 6). The results further supported the involvement of fibrillarin in an anti-PLPV response.

### 2.4. Fibrillarin Expression Levels Are Not Affected by PLPV Infection

In view of the observed effects of fibrillarin on PLPV accumulation, we decided to assess whether the expression levels of this host factor could be misregulated by the viral infection. To examine this issue, RT-qPCR was carried out to compare fibrillarin mRNA accumulation in systemic leaves (at 30 dpi) of mock-inoculated and infected wt *N. benthamiana* plants. The results showed that fibrillarin expression levels were not significantly modified in PLPV-infected tissues, at least, at any of the tested stages of infection (Figure 7). Hence, the presence of the virus does not result in transcriptional modulation of this major nucleolar host factor.

## 3. Discussion

In this work, the potential involvement of fibrillarin and coilin, signature proteins of the nucleolus and Cajal bodies, respectively, on PLPV infection was assessed. This study was prompted by the previous detection of PLPV VSR, protein p37, in these subnuclear compartments [16] and by the reported influence of such host factors in several plant–virus interactions [9,10,11,12].

The results showed that PLPV p37 interacts with both cellular proteins, an interaction that could be detected in the major accumulation sites of these polypeptides, nucleolus and CBs, as well as in the nucleoplasm (Figure 1). It has been postulated that the localization of proteins in the nucleolus and/or CBs, two membraneless nuclear organelles, is likely driven by protein–protein and/or protein–RNA interactions [25]. The obtained data suggest that fibrillarin and/or coilin could mediate retention of p37 in these organelles. though the participation of other components cannot be excluded. In any case, employment of *N. benthamiana* RNAi lines with reduced content of fibrillarin or coilin, have indicated that the detected interactions may have a differential impact on PLPV biological cycle. Knockdown expression of coilin had no noticeable effect on PLPV accumulation, whereas the decreased expression of fibrillarin resulted in an augmentation of PLPV titres. This augmentation was particularly evident or significant when plants were challenged with a VSR-defective PLPV. This observation is puzzling and suggests either that fibrillarin is a positive modulator of RNA silencing or, alternatively, it is involved in another antiviral response, which is also targeted by PLPV p37. Such targeting was further supported by the detected interaction between p37 and fibrillarin, an interaction that was lost in the p37 variant (p37WA) of the VSR-defective virus (Figure 3). Nevertheless, overexpression of fibrillarin resulted in diminished accumulation of wt PLPV (Figure 6), suggesting that p37 amounts produced during infection are insufficient to effectively cope with an overabundance of antiviral component(s). This situation would be reminiscent of plant viral resistance induced by transgenic expression of RNA silencing components [26].

Interactions between fibrillarin and a series of proteins encoded by both plant and animal viruses with an apparent effect on viral progression have been previously reported [9,12,27,28,29]. However, it is not always clear whether such an effect is the result of either subversion or interference with fibrillarin functions. In this context, it is worth mentioning that fibrillarin is a highly conserved 2′-O-methyltransferase responsible for methylation of rRNA and histone H2A in nucleoli, though increasing evidence supports its involvement in other processes, including rRNA processing, cell cycle control, nuclear shape, and cellular stress [30]. In addition, its role in bacterial resistance has been recently documented [31], and a potential participation in antiviral defences was also proposed [32]. Our results with PLPV would add a new example in this latter direction, suggesting that the plethora of fibrillarin activities is awaiting to be fully elucidated, reinforcing the connection.

Concerning coilin, proviral or antiviral roles, depending on the virus, have been advanced for it through the employment of the COILi *N. benthamiana* line used in the present work. Though the molecular mechanisms underlying these roles are still largely unknown, research with *Tobacco rattle virus* (TRV) suggested that antiviral functions could be related with a salicylic acid-mediated plant defence response which may be targeted by the TRV VSR, protein 16K [21]. Despite the fact that PLPV p37 has the ability to interact with and relocalize coilin to the nucleolus, similar to that described for TRV 16K, coilin reduction had no obvious consequences for PLPV infection. As found for PLPV, a coilin decrease seemed to have no repercussions on the *Potato virus X* infection [11], highlighting a strong virus-specific effect of some cellular factors, in line with that inferred from distinct studies with a variety of experimental approaches [33].

To conclude, from this and previous studies [16,19], we can envision a model in which PLPV p37 enters the nucleus through an importin alpha-mediated pathway and reaches high accumulation in Cajal bodies and nucleolus, most likely through its interactions with the hallmark proteins coilin and fibrillarin. The latter interaction interferes with antiviral plant defences either related to RNA silencing or to a yet undetermined pathway, which is also undermined by p37 (Figure 8). This scenario is also consistent with the detrimental consequences that impairment of p37 nuclear import and nucleolar localization have on PLPV infection [19]. Further investigations may shed light on the plant antiviral responses in which fibrillarin may take part.

## 4. Materials and Methods

### 4.1. DNA Constructs

Infectious PLPV clones consisting in binary plasmids (pMOG800-based constructs) containing a wt PLPV cDNA or a mutated PLPV cDNA (PLPV-mutp37WA), flanked by the *Cauliflower mosaic virus* 35S promoter and the terminator sequence of the *Solanum tuberosum* proteinase inhibitor II gene, were employed to inoculate plants [14,16]. The mutated cDNA harboured three nucleotide replacements in the p37 gene leading to an amino acid substitution (W28A) that, as reported previously [16], affects a GW motif in p37 that is critical for its VSR function but not for its packaging function. pROK2-sYFPN and pROK2-sYFPC-based plasmids directing expression of the p37 (wt and mutant p37WA) fused to the N- and C-terminal halves of the sYFP (aa 1 to 154 and 155 to 238, respectively) have been described previously [16]. The same plasmids were used as backbones to generate constructs directing expression of *N. benthamiana* fibrillarin (accession no. AM269909.1) and coilin (accession no. MK903618.1). Specific oligonucleotide primers (Table 1) were used to amplify the corresponding full-length ORFs through RT-PCR with SuperScript III One-Step RT-PCR System (Thermo Fisher Scientific) and total RNA extracts from *N. benthamiana* as templates. The primers included appropriate restriction sites to facilitate cloning of the amplified cDNA into pROK2-sYFPN and pROK2-sYFPC plasmids. A pROK2-based construct for expression of fibrillarin fused to monomeric red fluorescent protein (FIB:mRFP) and constructs for expression of transcription factors FUL and SOC fused to sYFP halves have been described elsewhere [9,20]. pMOG800-based plasmids directing expression of HA-tagged wt p37 and mutant p37WA have been described earlier [16].

The nucleotide sequence of all DNA constructs was verified with an ABI PRISM DNA sequencer 377 (Perkin-Elmer).

### 4.2. Agrobacterium Tumefaciens Transformation and Agroinfiltration of N. benthamiana Plants

Binary plasmids were used to transform *A. tumefaciens* strain C58C1 CH32 by the freeze/thaw shock method. Cultures of the transformed *A. tumefaciens* strains were infiltrated at an OD_660_ of 0.5 on the abaxial side of *N. benthamiana* leaves (two leaves per plant) using a 20 mL needleless syringe.

For PLPV agroinoculation, different lines of *N. benthamiana* plants were employed: (i) wt *N. benthamiana*, (ii) *N. benthamiana* transgenic plants in which fibrillarin gene was downregulated by expressing a hairpin construct (line FIBi) [10], and (iii) *N. benthamiana* transgenic plants in which coilin gene was similarly downregulated by expressing a hairpin construct (line COILi) [11]. Batches of 6–8 plants per line were used in each trial, and three experimental replicates were performed. Mock-inoculated plants served as negative controls.

For BiFC assays, wt *N. benthamiana* leaves were co-infiltrated with proper combinations of bacterial cultures and transformed with pROK2-based constructs that were mixed (OD_660_ = 0.5) prior to infiltration.

All plants were grown until four-to-six leaf (for PLPV inoculation) or six-to-eight leaf (for BiFC assays) stage in a greenhouse and kept under greenhouse conditions (16 h days at 24 °C, 8 h nights at 20 °C) until harvesting.

### 4.3. Confocal Microscopy

Reconstituted sYFP in BiFC assays was monitored in epidermal cells of *N. benthamiana*-infiltrated tissue at 72 h post-infiltration using a Leica TCS SL confocal microscope with an HCX PL APO ×40/1.25–0.75 oil CS objective. sYFP fluorescence was recorded by excitation with 488 nm argon laser line with emission being collected through a bandpass filter from 505 to 550 nm. For mRFP visualization, excitation was performed by means of a 543-nm green-neon laser line, and fluorescence emission was collected at 610 to 630 nm.

### 4.4. Co-Immunoprecipitation Assays

*N. benthamiana* leaves were agroinfiltrated with the construct directing expression of FIB:mRFP together with that directing expression of HA-tagged wt p37 or p37WA. Agroinfiltrated tissue was collected at 3 dpif, crushed to a fine powder with liquid nitrogen, and homogenized in 4 mL/g IP buffer (50-mM Tris–HCl, pH 7.5, 150-mM NaCl, 5% [*v/v*] glycerol, 0.1% [*v/v*] Triton, 1-mM MgCl_2_, 10-mM DTT, 2% polyvinylpolypyrrolidone (PVPP), 0.5-mM phenylmethylsulfonyl fluoride (PMSF), 1 µg/mL leupeptin, 1 µg/mL aprotonin and one tablet of complete proteinase inhibitor cocktail (Roche Life Science)) [34]. Cell debris were removed by centrifugation at 12,000× *g* for 15 min at 4 °C twice, and the clarified lysates were incubated with Proteintech ChromoTech RFP-Trap Agarose for 2 h with mild rotation. Beads were recovered by centrifugation at 500× *g* and washed four times with IP buffer without PVPP [34]. Proteins were eluted of 2X protein loading buffer (1.25 M Tris, pH 6.8, 10% SDS, 80% glycerol, 10% β-mercaptoethanol, and 0.02% bromophenol blue) after heating at 95 °C for 3 min. Western blot analysis of protein inputs and immunoprecipitates for detection of HA-tagged p37 molecules and FIB:mRFP was performed with HRP-conjugated anti-HA (Roche Applied Science) and anti-RFP (ChromoTech) antibodies, respectively. In the latter case, a HRP-conjugated sheep anti-mouse IgG (GE Healthcare) was employed as secondary antibody. Chemiluminescent signals were captured with ImageQuant 800 (Amersham).

### 4.5. RNA Extraction and Northern Blot Hybridization

Total RNA preparations from *N. benthamiana* leaves were obtained by phenol extraction and lithium precipitation [35]. For Northern blot analysis, 4 μg total RNA was denatured by glyoxal-dimethyl sulfoxide treatment, electrophoresed in 1% agarose gels, and blotted to nylon membranes (Hybond N+; GE Healthcare). After UV-crosslinking, membranes were incubated at 70 °C, in the presence of 50% formamide, with a ^32^P-radioactive RNA probe for detection of PLPV RNAs. The probe was generated by in vitro transcription of a pBluescript KS(+)-based construct containing the PLPV p37 gene (nt 2621–3637 of PLPV genome). After hybridization, the membranes were washed at room temperature three times (10 min each) in 2 × SSC plus 0.1% SDS and once at 55 °C in 0.1 × SSC plus 0.1% SDS. Hybridization signals were recorded though autoradiography.

### 4.6. RT-qPCR

Total RNA preparations from *N. benthamiana* plants with a RIN (RNA integrity number, Agilent) equal to or greater than 7 were treated with Turbo DNase (ThermoFisher). RT-qPCR was employed to compare either viral titres or fibrillarin expression levels among distinct samples. After DNase treatment, RNA preparations were subjected to reverse transcription (1 μg per reaction) with a Transcriptor high fidelity cDNA synthesis kit (Roche) using either random hexamer primers (to generate cDNAs for subsequent PCR amplification of PP2A gene fragment, employed as internal control, and of a virus genome fragment) or a combination of random hexamer primers and anchored-oligo(dT)_18_ primer (to generate cDNAs for subsequent PCR amplification of protein phosphatase 2A-PP2A-gene fragment and of fibrillarin gene fragment). The design of the primers for PCR was performed with Primer3 software (4 November 2017), developed by Rozen and Skaletsky, Whitehead Institute for Biomedical Research (Cambridge, MA, USA), using the following criteria: melting temperature ranging from 50 °C to 60 °C, PCR amplicon lengths of 100 to 200 bp, length of primer sequences of 19 to 25 nucleotides, and guanine-cytosine content of 40% to 60% (Table 1). Master mix for qPCR was prepared with 5× PyroTaq EvaGreen qPCR Mix Plus (ROX) (Cultek Molecular Bioline). Three biological replicates (with three technical replicates each) were performed for every type of sample. The PCR reactions were run and analysed using the QuantStudio^TM^ 3 (Applied Biosystems Inc., Life Technologies Corp.). Numerical data were expressed as mean ± standard deviation. To determine the statistical significance between the biological samples, a Student’s homoscedastic two-tailed *t*-test was used, considering significant differences as those with *p* value < 0.05.

## Figures and Tables

**Figure 1 plants-11-01903-f001:**
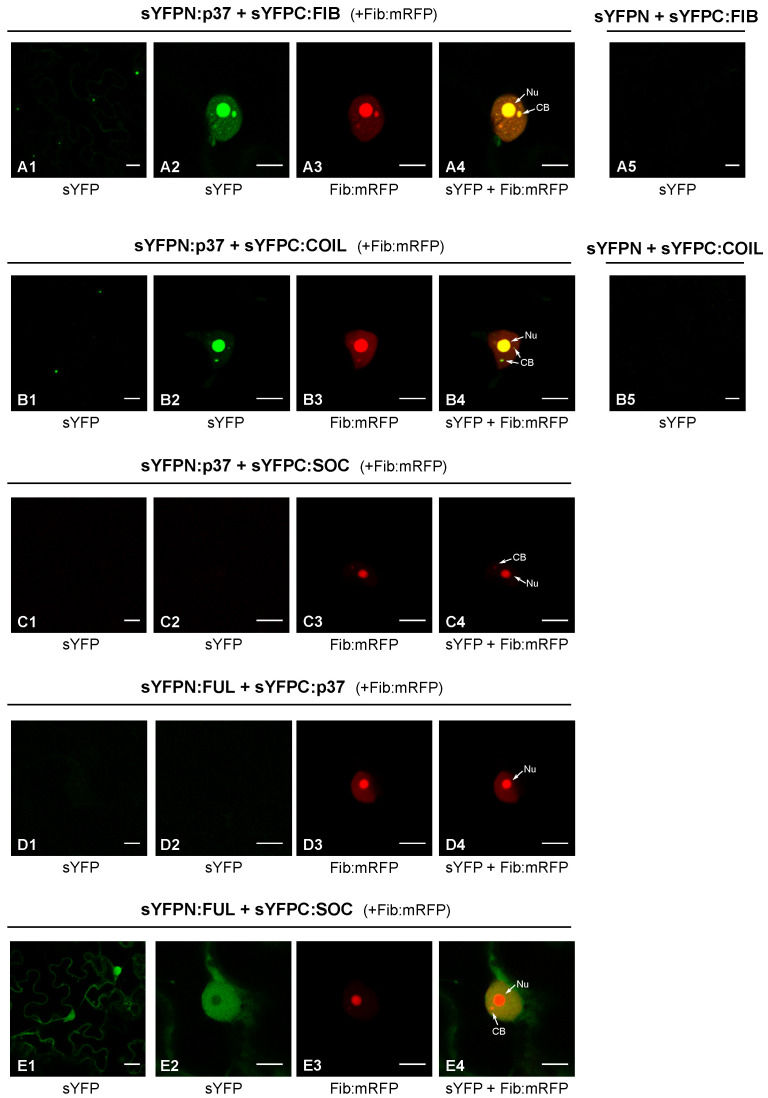
In vivo analysis of p37 interaction with fibrillarin and coilin. WT p37, fibrillarin (FIB), and coilin (COIL) were tagged at their N-terminus with YFP halves (sYFPN and sYFPC) and transiently co-expressed in *N. benthamiana* leaves to study the protein–protein interactions through a BiFC assay. FIB-mRFP was also co-expressed in these assays as a nucleolar/CB marker. Confocal laser scanning microscopy was used for the observation of fluorescence at 3 dpif. The micrographs (**A1**,**B1**) show a general view of sYFP-derived fluorescence in epidermal cells expressing sYFPN:FIB and sYFPN:COIL, respectively, in combination with sYFPC-tagged p37 molecules; the micrographs (**A2**,**B2**) show a close view of sYFP-derived fluorescence from the same protein-tagged combinations; the micrographs (**A3**,**B3**) correspond to the same close views but record the mRFP fluorescence derived from FIB-mRFP; the micrographs (**A4**,**B4**) show overlays of (**A2**,**A3**) and of (**B2**,**B3**), respectively. Equivalent images were obtained with reverse combinations (sYFPC:FIB or sYFPC:COIL co-expressed with sYFPN-tagged p37 molecule) (data not shown). Negative control combinations (sYFPN-sYFPC:FIB; sYFPN-sYFPC:COIL) are displayed in micrographs (**A5**,**B5**), respectively. Additional negative controls were included and shown in micrographs (**C1**–**C4**), (**D1**–**D4**), and (**E1**–**E4**). These controls corresponded to two transcription factors, FUL and SOC, that interact between them in the nucleus and cytoplasm (**E1**–**E4**) but that did not show interaction with p37 (**C1**–**C4**,**D1**–**D4**). Nu: Nucleolus. CB: Cajal body. The inset scale bar corresponds to 20 µm in the first column panels and to 10 µm in all remaining panels.

**Figure 2 plants-11-01903-f002:**
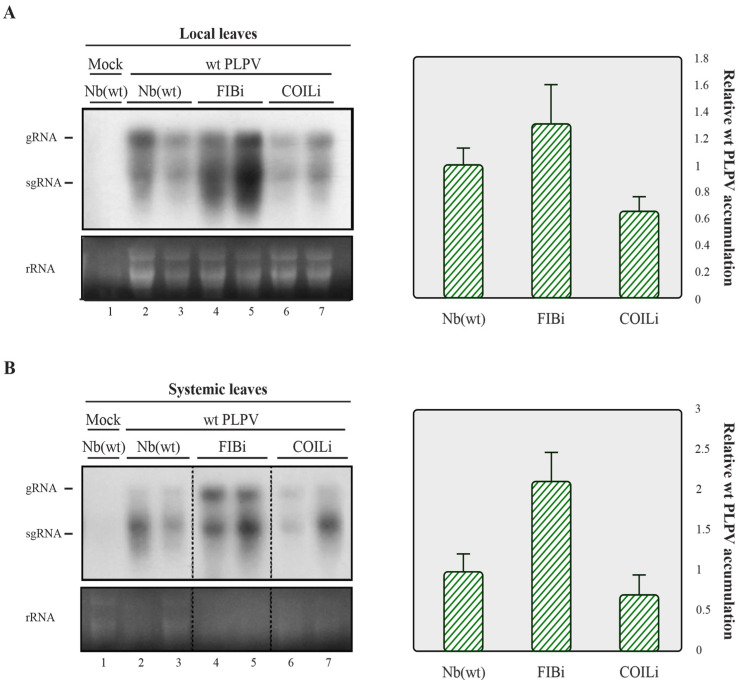
Accumulation of wt PLPV in lines of *N. benthamiana* with downregulated fibrillarin or coilin content. The wt PLPV was agroinoculated in wt and in FIBi and COILi *N. benthamiana* lines. Northern blot (left) and RT-qPCR (right) analyses of local (**A)** and systemic (**B**) leaves collected at 7 dpi and 30 dpi, respectively. For Northern blot, a PLPV- specific riboprobe was employed. Positions of the PLPV genomic (g) and subgenomic (sg) RNAs are indicated at the margin of the blots, and ethidium bromide staining of rRNAs is included below the blots as loading controls. Two distinct samples are shown for each virus–plant line combination. A mock-inoculated plant was included as negative control. In the graphs to the right, corresponding to the RT-qPCR measurement of relative PLPV accumulation, the bars depict the standard deviation from three independent biological replicates. The statistical significance was evaluated using a paired t-test (no significant differences were detected).

**Figure 3 plants-11-01903-f003:**
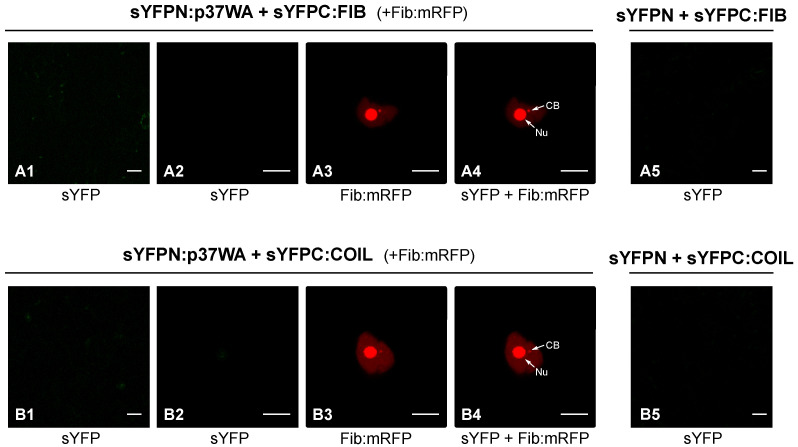
In vivo analysis of the potential interaction of p37WA mutant with fibrillarin and coilin. Protein p37WA, fibrillarin (FIB), and Coilin (COIL) were tagged at their N-terminus with YFP halves (sYFPN and sYFPC) and transiently co-expressed in *N. benthamiana* leaves to study the protein–protein interactions through a BiFC assay. FIB-mRFP was also co-expressed in these assays as a nucleolar/CB marker. Confocal laser scanning microscopy was used for the observation of fluorescence at 3 dpi. The micrographs (**A1**,**B1**) show a general view of YFP-derived fluorescence in epidermal cells expressing sYFPN:FIB and sYFPN:COIL, respectively, in combination with sYFPC-tagged p37WA molecules; the micrographs (**A2**,**B2**) show a close view of YFP-derived fluorescence from the same protein-tagged combinations; the micrographs (**A3**,**B3**) correspond to the same close views but record the RFP fluorescence derived from FIB:mRFP; the micrographs (**A4**,**B4**) show overlays of (**A2,A3,B2,B3**), respectively. Equivalent images were obtained with reverse combinations (sYFPC:FIB or sYFPC:COIL co-expressed with sYFPN-tagged p37WA molecule) (data not shown). Negative control combinations (sYFPN-sYFPC:FIB; sYFPN-sYFPC:COIL) are displayed in micrographs (**A5**,**B5**), respectively. Nu: nucleolus. CB: Cajal body. The inset scale bar corresponds to 20 µm in the first column panels and to 10 µm in all remaining panels.

**Figure 4 plants-11-01903-f004:**
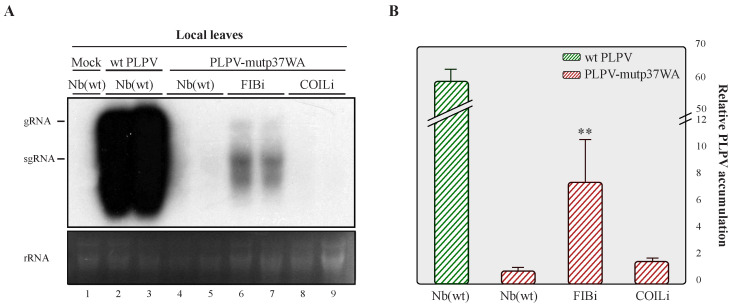
Accumulation of PLPV-mutp37GA in lines of *N. benthamiana* with downregulated fibrillarin or coilin content. The PLPV-mutp37WA was inoculated in wt, FIBi, and COILi *N. benthamiana* lines. Northern blot (**A**) and RT-qPCR (**B**) analyses of local leaves collected at 7 dpi. For Northern blot (**A**), a PLPV- specific riboprobe was employed. Positions of the PLPV genomic (g) and subgenomic (sg) RNAs are indicated at the left of the blot, and ethidium bromide staining of rRNAs is included below the blot as loading control. Two distinct samples are shown for each virus–plant line combination. A mock-inoculated plant was included as negative control. In (**B**), the bars depict the standard deviation from three independent biological replicates. The statistical significance was checked using a paired *t*-test (** *p* < 0.01).

**Figure 5 plants-11-01903-f005:**
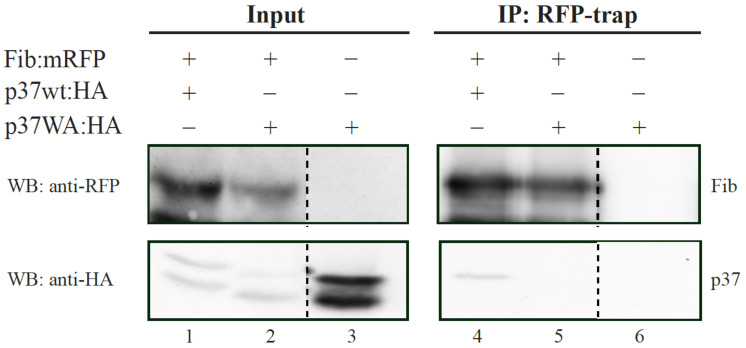
Analysis of fibrillarin interaction with wt p37 and mutant p37WA through a co-immunoprecipitation assay. FIB:mRFP was co-expressed with HA-tagged wt p37 or p37WA in *N. benthamiana* leaves. *N. bentamiana* leaves expressing only HA-tagged p37WA were included as control. Input protein extracts or immunoprecipitates (IP) obtained with the employment of ChromoTek RFP-Trap agarose were subjected to Western blot (WB) analysis using either an anti-RFP antibody (for detection of fibrillarin; upper blot) or anti-HA antibody (for detection of p37 molecule; lower blot).

**Figure 6 plants-11-01903-f006:**
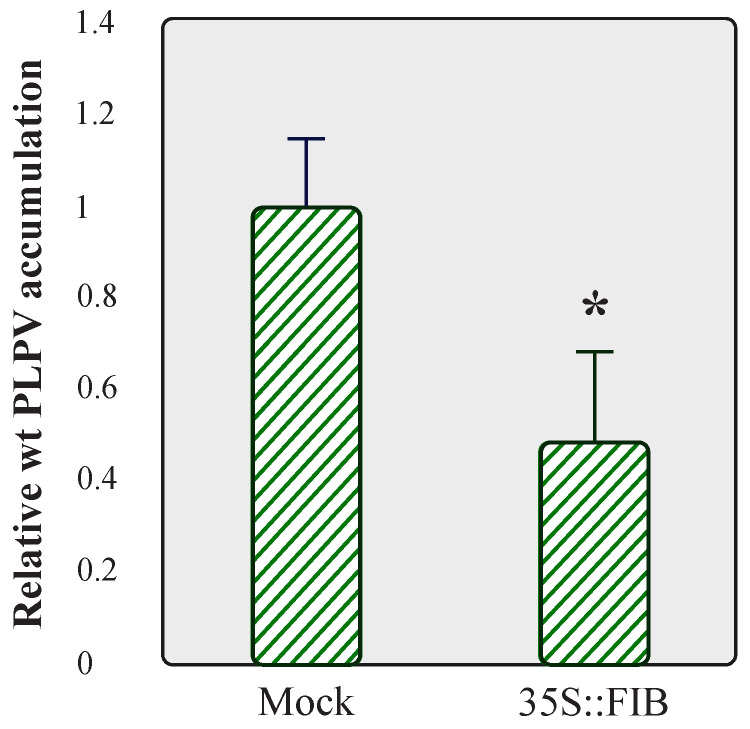
Analysis of PLPV accumulation in *N. benthamiana* leaves overexpressing fibrillarin. *N. benthamiana* leaves were agroinfiltrated with a pROK2-based plasmid directing expression of FIB:mRFP or with a plasmid without insert (mock control), and at 3 dpif, the leaves were agroinoculated with wt PLPV. Total RNA preparations were obtained at 3 dpi and subjected to RT-qPCR to estimate relative viral accumulation. The bars depict the standard deviation from three independent biological replicates. The statistical significance was evaluated using a paired *t*-test (* *p* < 0.05).

**Figure 7 plants-11-01903-f007:**
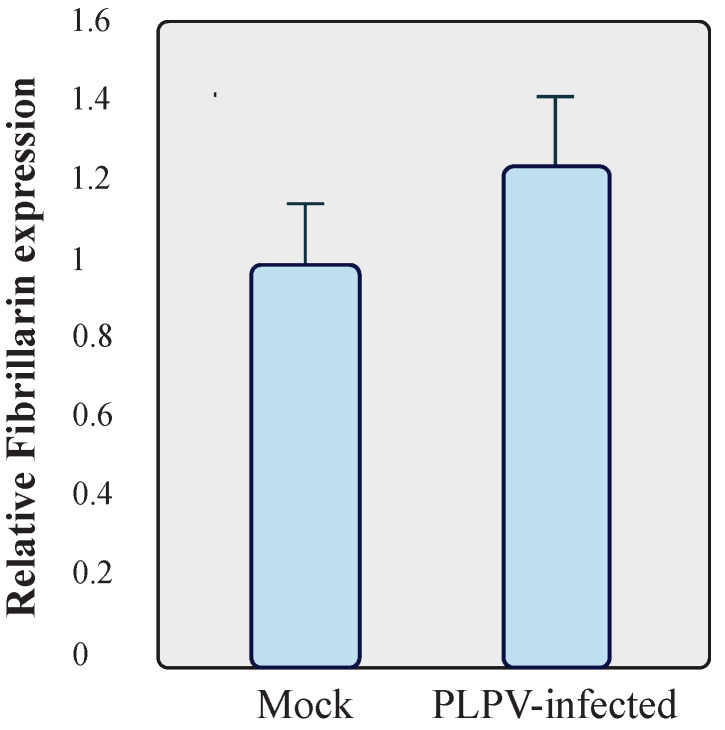
Assessment of mRNA fibrillarin levels. Total RNA preparations were obtained from mock and PLPV-infected wt *N. benthamiana* plants and subjected to RT-qPCR to estimate relative levels of fibrillarin transcripts. The bars depict the standard deviation from three independent biological replicates. The statistical significance was evaluated using a paired *t*-test (no significant differences were detected).

**Figure 8 plants-11-01903-f008:**
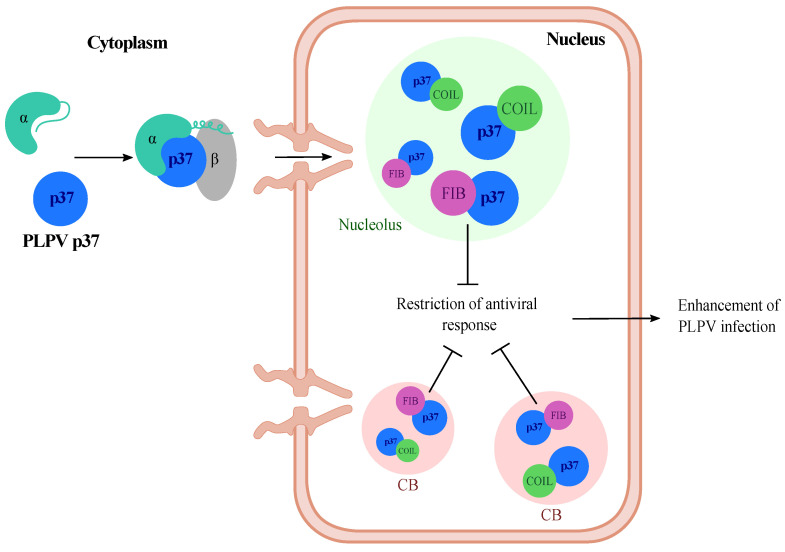
Proposed model reflecting the involvement of fibrillarin in an antiviral pathway targeted by PLPV p37. Protein p37 enters the nucleus assisted by an importin (α, β)-mediated pathway (Pérez-Cañamás and Hernández, 2018) and reaches high accumulation at nucleolus and Cajal bodies (CB). In these subnuclear structures, p37 interacts with the signature proteins, coilin and fibrillarin. The latter interaction interferes with a fibrillarin-linked antiviral pathway, thus favouring PLPV infection.

**Table 1 plants-11-01903-t001:** List of primers used in this work.

Gene	Primer	Position ^a^	Genbank Acc. No. ^b^	Sequence ^c^		Constructs / RT-qPCR
FIB	CH674 CH675	1–22 (S) 924–945 (AS)	AM269909.1	5′-GTGGATCCATGGTTGCACCAACTAGAGGTC-3′ 5′-GGGAGCTCTAGGCAGCAGCCTTCTGCTTC-3′	(*Bam*HI) *(Sac*I)	sYFPN:FIB sYFPC:FIB
COIL	CH712 CH713	142–161 (S) 2559–2583 (AS)	MK903618.1	5′-CAGGATCCATGGAGGGCGTTAGGCTTC -3′ 5′-CCGGTACCTCAAATTTTGTTCTGGGATCTTAG-3′	(*Bam*HI) (*Kpn*I)	sYFPN:COIL sYFPC:COIL
PLPV p27	CH718 CH719	86–105 (S) 167–185 (AS)	EU835946.1	5′-CGCTCCTCGGTCCTAACTTG -3′ 5′-ATTTTGGCCAACCCATGGA -3′		RT-qPCR
FIB	CH937 CH938	299–318 (S) 468–487 (AS)	AM269909.1	5′-ATTTGGTGCCTGGTGAAGCT-3′ 5′-TTCCTGATGCAGCTCCAAGG -3′		RT-qPCR
PP2A	CH436 CH437	996–1016 (S) 1075–1095 (AS)	MF996339.1	5′-ACTTGGTGCCCTTTGTATGC-3′ 5′-TGGACCAAATTCTTCTGCAA-3′		RT-qPCR

^a^ Positions covered by the primers in the corresponding genes. (S) and (AS): sense and antisense. ^b^ Genbank accession number of the genes used in the analysis. ^c^ Engineered restriction sites for cloning purposes are underlined.

## Data Availability

Not applicable.

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
