# Peer review of "A Viral Suppressor of RNA Silencing May Be Targeting a Plant Defence Pathway Involving Fibrillarin"

_plants, 2022, doi:10.3390/plants11151903_

Round 1

Reviewer 1 Report

The manuscript by Perez-Canamas et. al. supports an antiviral role for fibrillarin against PLPV. The authors also provide evidence that p37 counteracts fibrillarin activities through direct interactions. The data presented is sound and the conclusions are justified. However, the antiviral activity of Fibrillarin appears modest and could be further supported by over-expressing Fibrillarin to determine if wt PLPV levels can be substantially reduced. Together, this research highlights the promiscuous activities of fibrillarin and will be of great interest to virologists.

Major:

Figure 2 and Figure 4: The authors conclude that no significant change in WT PLPV levels was observed in FIBi lines. However, this reviewer sees a significant increase in wt PLPV levels based on the Northern blot presented in Figure 2. The gRNA levels appear to be >2-fold higher and the RT-qPCR data appears significant, albeit much less significant compared to the data presented in Fig. 4.

The Knock-down efficiencies of the FIBi and COILi lines should be determined using RT-qPCR.

To further support the authors conclusion that Fibrillarin has antiviral activity towards PLPV, the following complementary assay could be performed. FIB can be over-expressed (35S promoter) to determine the extent that FIB can reduce PLPV accumulation.

Minor:

Line 103: should read "considered it relevant"

Figure 1: Change the label of mRFP to "Fib-mRFP" to make clear this is a fibrillarin marker and not free mRFP.

Figure 1 Legend: How is the scale bar 20 um for all images when A1 and B1 are a wide view? Also, a typical nuclei tends to be around 10 um, and based on the images in Figure 1, the nuclei are >20 um in diameter. Please confirm the error bar parameters.

Line 147: A reference that demonstrates that coilin is only found in CBs and not the nucleolus is warranted. Is nucleolar localization of Coilin an artifact of 35S-driven overexpression?

Fig. 3: What controls were in-place to ensure p37WA was expressed? A western blot confirming p37WA expression is important to demonstrate p37 was expressed and did not become unstable due to the mutation. Scale bars again seem incorrect when comparing panels 1 and 2.

Line 181: a space between aminoacid is needed

Line 207: should read "paralleling what has been reported in N. benthamiana..."

Author Response

This short paper is aimed at detection of PLPV p37 interaction with nucleolar proteins. The authors claim that they have demonstrated the p37 interaction with fibrillarin and show that PLPV with disabled silencing suppression function is capable of amplification in fibrillarin RNAi plants.

The presented data are promising, however the p37 interaction with fibrillarin requires verification by use of other methods and important controls are missing.

To my mind, additional experimental data should be included into the manuscript before it can be reconsidered for publication. 

Major concerns

  1. Problems with BiFC experiments.

  1. a) Generally, a positive signal in BiFC experiments is not sufficient to state that analyzed proteins interact, and this conclusion should be supported by other methods. 

  1. b) Besides, the BiFC shown in Fig. 1 was carried out without any negative control such as sYFPN fused to an unrelated protein with a similar localization or, most relevant, sYFPN carrying a nucleolar localization signal.

  1. c) The BiFC data in Fig. 3 are presented without any positive control that would ensure that the sYFPC-p37WA construct is able to produce fluorescence under BiFC conditions. sYFPN-p37WA can be used. Without such control the statement that p37WA ‘lost the nucleolar/CB localization’ is incorrect, as the localization of p37WA remains unknown. 

Answer: As shown in Pérez-Cañamás and Hernández (2015), p37WA is perfectly able to produce fluorescence under BiFC conditions. In Fig. 6 and 7 of that paper, it is shown (through BiFC approach) that the mutant protein is able to self-interact and, also, to interact (though weakly compared to wt p37) with AGO1 and AGO4. Moreover, the distribution pattern of that protein was also addressed showing the loss of the nucleolar localization caused by the engineered mutation (Fig. 5 of that paper). We have included the remark “as shown previously” in the main text (lines 196-198) to make this point clearer.

On the other side, new BiFC controls, corresponding to transcription factors, have been included in Figure 1 (panels C-E), further validating the detected interactions.

Finally, interaction of wt p37 with fibrillarin has been corroborated by Co-IP (new Figure 6).

  1. Fig. 4: As the effect of fibrillarin silencing on virus amplification was detected for PLPV-mutp37WA, the influence of this particular mutant, rather than the wt virus, on the fibrillarin expression level should be presented. 

Answer: This mutant virus is unable to establish a productive infection. Its accumulation levels are barely detectable, as shown in Fig. 4 and in previous papers (Pérez-Cañmás and Hernández, 2015; Pérez-Cañamás et al., 2021), unless it is bioassayed in mutant plants (e.g., FIBi –this work-, DCL or AGO mutants –Pérez-Cañamás et al., 2021- ). In these condition, we think that the suggested assay may be useless.

  1. The English should be corrected by a native speaker or professional language service. Done.

Minor points

- Lines 69-79: Excessively long explanation of silencing - not relevant in the context of this paper. 

Answer: The text about this issue has been shortened as requested.

- Fig.1: Panels A1 and B1 (lower magnification images) are misleading as the legend indicates that the scale bar is ‘20 um in all panels’. I suggest deleting panels A1 and B1 as these are unnecessary.

Answer: We think that panels A1 and B1 give an overview of fluorescence patterns. Scale bar has been corrected.

Reviewer 2 Report

This short paper is aimed at detection of PLPV p37 interaction with nucleolar proteins. The authors claim that they have demonstrated the p37 interaction with fibrillarin and show that PLPV with disabled silencing suppression function is capable of amplification in fibrillarin RNAi plants.

The presented data are promising, however the p37 interaction with fibrillarin requires verification by use of other methods and important controls are missing.

To my mind, additional experimental data should be included into the manuscript before it can be reconsidered for publication. 

Major concerns

1. Problems with BiFC experiments.

a) Generally, a positive signal in BiFC experiments is not sufficient to state that analyzed proteins interact, and this conclusion should be supported by other methods. 

b) Besides, the BiFC shown in Fig. 1 was carried out without any negative control such as sYFPN fused to an unrelated protein with a similar localization or, most relevant, sYFPN carrying a nucleolar localization signal.

c) The BiFC data in Fig. 3 are presented without any positive control that would ensure that the sYFPC-p37WA construct is able to produce fluorescence under BiFC conditions. sYFPN-p37WA can be used. Without such control the statement that p37WA ‘lost the nucleolar/CB localization’ is incorrect, as the localization of p37WA remains unknown. 

2. Fig. 4: As the effect of fibrillarin silencing on virus amplification was detected for PLPV-mutp37WA, the influence of this particular mutant, rather than the wt virus, on the fibrillarin expression level should be presented. 

3. The English should be corrected by a native speaker or professional language service.

Minor points

Lines 69-79: Excessively long explanation of silencing - not relevant in the context of this paper. 

Fig.1: Panels A1 and B1 (lower magnification images) are misleading as the legend indicates that the scale bar is ‘20 um in all panels’. I suggest deleting panels A1 and B1 as these are unnecessary.

Author Response

Comments and Suggestions for Authors

This short paper is aimed at detection of PLPV p37 interaction with nucleolar proteins. The authors claim that they have demonstrated the p37 interaction with fibrillarin and show that PLPV with disabled silencing suppression function is capable of amplification in fibrillarin RNAi plants.

The presented data are promising, however the p37 interaction with fibrillarin requires verification by use of other methods and important controls are missing.

To my mind, additional experimental data should be included into the manuscript before it can be reconsidered for publication. 

Major concerns

  1. Problems with BiFC experiments.

  1. a) Generally, a positive signal in BiFC experiments is not sufficient to state that analyzed proteins interact, and this conclusion should be supported by other methods. 

  1. b) Besides, the BiFC shown in Fig. 1 was carried out without any negative control such as sYFPN fused to an unrelated protein with a similar localization or, most relevant, sYFPN carrying a nucleolar localization signal.

  1. c) The BiFC data in Fig. 3 are presented without any positive control that would ensure that the sYFPC-p37WA construct is able to produce fluorescence under BiFC conditions. sYFPN-p37WA can be used. Without such control the statement that p37WA ‘lost the nucleolar/CB localization’ is incorrect, as the localization of p37WA remains unknown. 

Answer: As shown in Pérez-Cañamás and Hernández (2015), p37WA is perfectly able to produce fluorescence under BiFC conditions. In Fig. 6 and 7 of that paper, it is shown (through BiFC approach) that the mutant protein is able to self-interact and, also, to interact (though weakly compared to wt p37) with AGO1 and AGO4. Moreover, the distribution pattern of that protein was also addressed showing the loss of the nucleolar localization caused by the engineered mutation (Fig. 5 of that paper). We have included the remark “as shown previously” in the main text (lines 196-198) to make this point clearer.

On the other side, new BiFC controls, corresponding to transcription factors, have been included in Figure 1 (panels C-E), further validating the detected interactions.

Finally, interaction of wt p37 with fibrillarin has been corroborated by Co-IP (new Figure 6).

  1. Fig. 4: As the effect of fibrillarin silencing on virus amplification was detected for PLPV-mutp37WA, the influence of this particular mutant, rather than the wt virus, on the fibrillarin expression level should be presented. 

Answer: This mutant virus is unable to establish a productive infection. Its accumulation levels are barely detectable, as shown in Fig. 4 and in previous papers (Pérez-Cañmás and Hernández, 2015; Pérez-Cañamás et al., 2021), unless it is bioassayed in mutant plants (e.g., FIBi –this work-, DCL or AGO mutants –Pérez-Cañamás et al., 2021- ). In these condition, we think that the suggested assay may be useless.

  1. The English should be corrected by a native speaker or professional language service. Done.

Minor points

- Lines 69-79: Excessively long explanation of silencing - not relevant in the context of this paper. 

Answer: The text about this issue has been shortened as requested.

- Fig.1: Panels A1 and B1 (lower magnification images) are misleading as the legend indicates that the scale bar is ‘20 um in all panels’. I suggest deleting panels A1 and B1 as these are unnecessary.

Answer: We think that panels A1 and B1 give an overview of fluorescence patterns. Scale bar has been corrected.

Round 2

Reviewer 2 Report

The authors provided requested controls and improved the text, therefore the paper can be accepted for publication in present form.